# Postlarval Shrimp-Associated Microbiota and Underlying Ecological Processes over AHPND Progression

**DOI:** 10.3390/microorganisms13040720

**Published:** 2025-03-24

**Authors:** Zhongjiang Zhou, Jiaqi Lu, Pingping Zhan, Jinbo Xiong

**Affiliations:** 1State Key Laboratory for the Quality and Safety of Agro-Products, School of Marine Sciences, Ningbo University, Ningbo 315211, China; 2Key Laboratory of Aquacultural Biotechnology, Ministry of Education, School of Marine Sciences, Ningbo University, Ningbo 315211, China

**Keywords:** postlarval shrimp, progressed AHPND, driving lineages, network stability, biomarkers

## Abstract

Postlarval shrimp frequently face threats from acute hepatopancreatic necrosis disease (AHPND). Although AHPND affects both postlarval and adult shrimp, abiotic and biotic factors are distinct between life stages, such as rearing water nutrient levels and host life stage-dependent microbiota. The response of postlarvae-associated microbiota to AHPND, however, remains largely unexplored compared with its effects on juvenile and adult shrimp. To address this knowledge gap, a comparative analysis of postlarvae-associated microbiota and the ecological processes underlying AHPND progression was performed by sequencing the bacterial V3–V4 hypervariable region of the 16S rRNA gene. AHPND infection was validated by high copies of *pirAB* genes (Toxin 1) in diseased shrimp hepatopancreas. Advanced AHPND significantly altered the structure of the postlarvae-associated microbiota, with significant enrichment of Bacilli and Bdellovibrionia species in healthy larvae compared with matched AHPND-infected cohorts, although gut microbiota recovery was observed at the late disease stage, corresponding with the cessation of postlarval mortality. AHPND infection explained 11.0% (*p* < 0.001) of the variance in community structures, whereas postlarvae days post hatching also significantly influenced bacterial communities (7.1% variance, *p* < 0.001). AHPND-infected shrimp exhibited reduced homogeneous selection and increased dispersal limitation and drift governing their microbiota. These changes were primarily driven by specific microbial lineages, including enriched Bin36 Rhodobacteraceae and Bin11 Flavobacteriaceae, and suppressed Bin63 *Vibrio* and Bin9 *Bacillus* in AHPND-infected shrimp. After excluding shrimp age effect, 13 AHPND-discriminatory taxa were identified, accurately distinguishing infected shrimp from healthy individuals with 100% precision. Furthermore, AHPND outbreak weakened the network complexity and stability, which was driven by the suppressed keystone taxa that were positively associated with network robustness. Collectively, our findings deepen the understanding of the inextricable interplay between postlarval shrimp health, microbiota dynamics, and survival, as well as the underlying ecological mechanisms over AHPND progression.

## 1. Introduction

*Penaeus vannamei* is a widely aquacultured shrimp species worldwide, and the success of its aquaculture is heavily dependent on the quality of the postlarval or nursery phase [1]. Artificial propagation within hatcheries plays a crucial role in generating premium-grade shrimp seeds, ensuring a stable and consistent supply of postlarvae for large-scale cultivation [2]. During their early life stages, shrimp are particularly susceptible to infectious pathogens, with acute hepatopancreatic necrosis disease (AHPND) posing a critical threat to postlarval individuals [3,4,5]. AHPND is caused by *Vibrio* strains harboring the pVA1 virulence-associated plasmid, which encodes the PirAB toxins responsible for inducing hepatopancreatic necrosis and epithelial cell sloughing in shrimp [6]. Besides the previously identified *Vibrio parahaemolyticus*, various *Vibrio* strains have been identified as pathogens responsible for AHPND, including *V. harveryi* [7], *V. owensii* [8], and *V. campbelli* [9]. AHPND is notorious for causing rapid and widespread mortality, presenting a significant challenge to the shrimp aquaculture sector [10,11]. Therefore, establishing effective strategies for diagnosing shrimp AHPND outbreaks is both essential and urgent, particularly for vulnerable postlarvae.

The host-associated microbiotas play crucial roles in maintaining essential functions such as digestion [12], immunity [13], and overall fitness, particularly during the early life stages [14]. Following hatching, the intestine, skin, and oral microbial communities of shrimp are gradually established [15]. Additionally, early colonizers can either facilitate or antagonize the colonization of later arrivals by modifying host physiology or immunity [16,17]. For these reasons, microbial colonization during the early life stage is considered particularly important due to its lasting impact on adult health. It is noteworthy that existing studies primarily focus on juvenile and adult shrimp affected by AHPND [18,19,20], largely neglecting their postlarval counterparts. Shrimp diseases can disrupt their ability to selectively recruit beneficial symbionts, resulting in a stochastic gut microbiota [21]. For instance, AHPND has been shown to induce dysbiosis in the gut microbiota of infected adult shrimp [22]. AHPND infection significantly reduces microbial diversity in the gut and hepatopancreas of juveniles but increases the abundance of γ-Proteobacteria, particularly members of the *Vibrio* genus, which are known opportunistic pathogens for shrimp [20]. Nevertheless, an elevated abundance of gut *Vibrio* populations has been observed in shrimp during larval and postlarval stages, with no apparent harm to the shrimp [23]. In contrast, postlarval shrimp infected with AHPND show no change or even a decrease in the abundance of *Vibrio* [24,25]. Indeed, the gut microbiotas are highly dynamic throughout shrimp development [14]. Ample evidence suggests that the responses of microbial communities to disturbances depend on their initial composition [26]. Thus, it is likely that postlarval and adult shrimp respond differently to AHPND infection. Specifically, postlarval shrimp experience massive mortality (up to 100%) within 20–30 days post-stocking, which is driven by immature immunity and toxin-mediated necrosis of midgut/hindgut epithelia, leading to rapid cell sloughing and nutrient absorption collapse [27]. In contrast, adult shrimp develop hepatopancreas-specific lesions (e.g., tubular necrosis, hemocyte infiltration) and exhibit delayed mortality due to robust immune defenses, although later stages are frequently complicated by secondary infections like vibriosis [28]. Despite the commercial and scientific importance of early-life stages, AHPND’s impact on postlarval shrimp microbiomes remains critically understudied compared to adults.

Community assembly theory provides a robust framework for understanding the complex processes that shape the structure and function of microbial communities [29,30]. The assembly of microbial communities is believed to be influenced by two interconnected mechanisms: deterministic processes, which are grounded in niche theory, and stochastic processes, which arise from neutral theory [31]. The Infer Community Assembly Mechanisms by phylogenetic bin-based null model (iCAMP) offers a sophisticated methodology for quantitatively assessing the ecological processes that dictate community dynamics. This approach focuses on phylogenetic bins of taxonomic groupings based on genetic similarities or their positions within the phylogenetic tree—rather than on the community as a whole. iCAMP demonstrates superior performance in terms of precision, sensitivity, specificity, and overall accuracy when compared to previous methodologies [32]. The relative roles of ecological processes may be altered by shrimp diseases. For instance, the importance of homogeneous selection diminishes in adult shrimp affected by advanced white feces syndrome (WFS), thereby reducing their colonization resistance to invading pathogens [33]. However, it remains unclear whether this pattern also applies to postlarval shrimp infected with AHPND. Furthermore, current research on shrimp microbiota primarily focuses on the impact of diseases on overall community ecological processes [15,21], often neglecting the specific taxa that drive these changes. Understanding these specific taxa could provide critical targets for microbial-based interventions [34]. Therefore, there is a compelling need to uncover the underlying ecological processes and the roles of specific bacterial taxa.

A significant research effort has demonstrated a causal association between dysbiosis in the shrimp gut microbiome and outbreaks of WFS [35]. Although the role of microbial dysbiosis as either a primary cause or a secondary effect of diseases remains a topic of debate, gut microbial biomarkers have been identified for differentiating shrimp health status. Previous studies have primarily relied on differential abundance analyses to identify potential biomarkers by comparing the microbial profiles of healthy versus diseased shrimp [36]. However, these methodologies are often limited by factors such as small sample sizes or the complex nature of diseases, leading to potentially unreliable or inconsistent results [37]. Furthermore, focusing solely on the differential abundance between host health statuses tends to narrow the analysis to individual taxa, which may overlook the broader interactions among community members [38]. In contrast, machine learning algorithms identify biomarkers based on their impact on disease outcomes, a process known as “feature selection” [38]. The interactions among shrimp gut commensals—including mutualistic, symbiotic, and competitive associations—are critical for establishing colonization resistance and determining overall shrimp health [39]. Therefore, it is essential to identify the keystone species that facilitate the transition from a healthy state to a diseased condition.

This study aimed to address several key aspects of shrimp-associated microbiota and its implications for their health: (i) to uncover how the microbiota of postlarval shrimp responds to advanced AHPND, (ii) to investigate the ecological processes and key microbial members driving postlarval-associated microbiota during AHPND progression, and (iii) to identify biomarkers for diagnosing AHPND infection.

## 2. Materials and Methods

### 2.1. Experimental Design and Sampling

*P. vannamei* postlarvae were enrolled from 14 tanks (3 tons) in a commercial hatchery. All tanks were subject to the same management protocols concerning rearing water, feeding, and water exchange rates. Water parameters remained relatively stable throughout the production cycle, with temperatures maintained between 32.5 °C and 33.0 °C, pH values ranging from 7.9 to 8.1, salinity at 25.0 g/L, and dissolved oxygen levels around 5.0 mg/L. At 11 days post hatching (dph), mortality was observed in a few tanks, with the presence of detectable *pirAB* genes in the affected shrimp. At 15 dph, typical disease signs of AHPND emerged, including erratic swimming patterns (intermittent hyperactivity followed by lethargy), reduced feeding activity, pallor, and atrophy hepatopancreas. AHPND infection was further validated by a high mortality rate and copies of *pirAB* genes (Toxin 1) in diseased shrimp hepatopancreas using an IQ REAL™ AHPND Quantitative kit (Gene Reach Biotechnology Corps, Shanghai, China) [11], while *pirAB* genes were undetectable in healthy individuals (Appendix A). The healthy postlarval shrimp were transferred to greenhouse ponds by day 23 dph. To investigate the impact of progressed AHPND on the associated microbiota of the postlarvae, we collected 56 samples from the tanks containing AHPND-infected shrimp, along with matched healthy controls at 11, 15, 18, and 21 dph, respectively (Appendix A). Due to the diminutive size of the postlarval shrimp, 1 g from each sample was washed twice using sterile 0.9% NaCl to eliminate transient microbes and subsequently stored at −80 °C until DNA extraction.

### 2.2. PCR Amplification and Amplicons Sequencing

Genomic DNA (gDNA) of the postlarval shrimp was extracted using the ZymoBIOMICS DNA Microprep isolation kit (Zymo Research, San Diego, CA, USA), following the manufacturer’s provided protocols. The final dataset comprised 47 valid samples after excluding those that failed DNA quality control. This included 22 healthy controls with a survival rate exceeding 80% and 25 AHPND-infected shrimp with a survival rate below 20% (Appendix A). The V3–V4 hyper-variable regions of bacterial 16S rRNA genes were amplified using the 341F (5′-CCTACGGGNBGCASCAG-3′) and 806R (3′-GACTACNVGGGTATCTAATCC-5′) paired primers, as described in previous studies [40]. Paired-end sequencing was performed on the Illumina Nova Seq 6000 P250 platform (Illumina, San Diego, CA, USA).

### 2.3. Amplicon Data Processing

The raw reads underwent analysis using the QIIME2 framework [41]. Specifically, the DADA2 algorithm was utilized to denoise the raw reads and assemble amplicon sequence variants (ASVs) [41]. The Vsearch plugin utilizing the UCHIME algorithm was used to identify and remove chimeric sequences by comparing input sequences against each other or a reference database. During quality filtering, reads shorter than 300 base pairs, those containing ambiguous bases, and those with a Phred quality score < 20 (Q20; 1% error probability per base) were excluded to minimize false ASV calls. The taxonomy of each ASV was annotated by blasting its representative sequence against the Silva 138 database [42]. In subsequent analyses, Archaea, chloroplasts, unclassified bacteria, and singletons were removed from the dataset. To ensure uniform sequencing depth across samples, reads were rarefied to 77, 110.

### 2.4. Construction of Diagnosis Model

To eliminate variations in bacterial communities associated with shrimp ontogeny (here was days post hatching), the abundance of ASVs was regressed against dph in healthy individuals using the “randomForest” package (Version 4.7-1.2). ASVs were ranked based on their importance in discriminating dph. The minimal set of dph-discriminatory ASVs that optimized classification accuracy was identified through 10-fold cross-validation. Subsequently, the top dph-discriminatory ASVs were removed from the bacterial communities, and AHPND-discriminatory ASVs that differentiated AHPND-affected shrimp from healthy individuals were screened, utilizing data from 47 enrolled samples (Appendix A). The abundance of these AHPND-discriminatory ASVs, along with the mean decrease in accuracy, served as key indicators for classifying shrimp into either the “Health” or “AHPND” category. The diagnostic model was developed using 32 samples that were randomly selected from the 47 subjects (Appendix A), with the remaining 15 samples serving as validation data. Model performance was assessed by comparing the actual health status with the diagnosed health status, establishing a correctness threshold of 50%.

### 2.5. Microbial Community Assembly Analysis

A phylogenetic bin-based null model (iCAMP) was applied to examine the ecological processes that govern postlarval shrimp-associated bacterial communities. The β Net Relatedness Index (βNRI) and the modified Raup–Crick metric were calculated within each bin to quantify the roles of ecological processes [32]. This method facilitated the assessment of the relative contributions of heterogeneous selection, homogeneous selection, homogenizing dispersal, dispersal limitation, and drift in shaping bacterial communities and various phylogenetic groups [43]. Here, we focused on how AHPND affected the contributions of these five processes. In iCAMP, “bins” were identified using a phylogenetic distance threshold of 0.2 [32]. Thus, ASVs with a pairwise phylogenetic distance less than 0.2 were clustered into the same bin, with node support assessed by SH-aLRT values at 95% confidence. Only branches with SH-aLRT values ≥ 0.95 are considered statistically significant [44]. The resulting data were visually represented and organized using the Interactive Tree Of Life (iTOL) platform.

### 2.6. Interspecies Interactions of Postlarvae-Associated Microbiota Between Health Status

To quantitatively compare the effect of AHPND on interspecies interactions among bacteria, co-occurrence networks were constructed for healthy and AHPND-infected cohorts, respectively. After removing of the dph-discriminatory ASVs, significant correlations (|r| > 0.6, adjusted *p* < 0.05) between ASVs were identified using the SparCC algorithm (1000 permutations) within FastSpar v1.0 [45]. The NetMoss score for each node was calculated to assess its importance in the network transition from healthy to AHPND-infected shrimp [46]. Several topological properties were evaluated using the “igraph” package in R. Networks were visualized with Gephi v0.10.0 [47]. Total cohesion was calculated by summing the absolute values of both positive and negative cohesion, reflecting network complexity [48]. Network robustness was measured by determining the proportion of ASVs retained when 50% of ASVs were randomly excluded from the network [49]. Network invulnerability was estimated by comparing changes in natural connectivity following the random removal of nodes [50]. A minor decrease in natural connectivity, indicated by a lower absolute regression slope, suggests greater network stability. The association between cohesion values, robustness, and the relative abundance of key ASVs during disease progression was assessed using linear regression.

### 2.7. Statistical Analysis

Constrained analysis of principal coordinates (CPA) was utilized to examine the contributions of dph and health status on community structures, employing Bray–Curtis distances [51]. Dissimilarity in bacterial communities across health statuses was analyzed using analysis of similarity (ANOSIM) for each sampling period. Differential taxa between shrimp health statuses were identified through LEfSe analysis, implemented in the “microbiomeMarker” package (Version 3.20) with the lefse function. The linear discriminant analysis (LDA) score indicates the degree of differences, with larger values signifying more significant differences [52]. In this study, taxa with LDA scores ≥ 4 and *p* < 0.05 were considered as differential taxa.

## 3. Results

### 3.1. Shrimp Mortality over AHPND Progression

In comparison to healthy controls, shrimp infected with AHPND displayed a significantly lower survival rate (Wilcoxon test, *p* < 0.01), with 17.9 ± 11.3% for AHPND-infected postlarvae versus 67.8 ± 7.5% for healthy controls. However, no further mortality was observed on 21 dph and thereafter (Appendix A).

### 3.2. Responses of Postlarvae-Associated Bacteria Community to AHPND

A total of 4,276,999 high-quality sequences were generated from the enrolled 47 samples. The sequencing depth ranged from 77,110 to 114,866, with an average of 91,000 ± 8619 (mean ± standard deviation), and the data were rarefied to 77,110 reads per sample for subsequent analyses. In healthy postlarvae, bacterial communities were predominantly composed of Alphaproteobacteria and Gammaproteobacteria, followed by Bacteroidia, Bacilli, and Bdellovibrionia (Appendix A). The relative abundance of Bacilli decreased linearly (*p* < 0.01) during healthy postlarvae development, whereas Bdellovibrionia exhibited an opposite trend (*p* < 0.01). In contrast, the abundance of Bacteroidia decreased linearly (*p* < 0.05) during AHPND progression. Furthermore, Bacilli and Bdellovibrionia species were significantly enriched in healthy postlarvae compared with matched AHPND-infected cohorts (Appendix A). Compared with AHPND-infected shrimp, the genera *Bacillus* and *Salipiger* were significantly more abundant in the healthy controls, whereas the genera *Gilvibacter* and *Spongiimonas* exhibited an opposite trend (Appendix A).

The α-diversity of AHPND-affected postlarvae was significantly lower than that of healthy postlarvae at 11 dph, with no significant differences (*p* > 0.05) observed at the subsequent stages (Figure 1A). The CPA biplot showed that bacterial community structures clustered by postlarval health status (Figure 1). However, there was no significant difference (ANOSIM *r* = 0.003, *p* = 0.49) in community structure between healthy and infected cohorts at 21 dph (Appendix A). This trend was more pronounced when comparing microbiota across health statuses at each sampling point (Appendix A). PERMANOVA revealed that health status explained 11.0% (*R*^2^ = 0.11, *p* < 0.001) of the variance in community structures, whereas postlarvae dph also significantly influenced bacterial communities (*R*^2^ = 0.071, *p* < 0.001) (Figure 1B).

### 3.3. Bacterial Lineages Featured Shrimp Health Status

To investigate the effects of AHPND on the microbiota associated with postlarvae, we screened the bacterial lineages that distinguished AHPND-infected individuals from healthy controls. Healthy shrimp consistently showed a higher abundance of Firmicutes, particularly *Bacillus* species, in comparison to matched AHPND subjects (Figure 2). Additionally, some differential lineages were stage specific. For instance, Gammaproteobacteria, Alteromonadaceae, and *Roseobacter* species were significantly enriched in healthy shrimp, whereas Caulobacterales and *Nautella* were more prevalent in AHPND-infected shrimp at 11 dph (Figure 2A). At 15 dph, healthy shrimp displayed notable enrichments of Bdellovibrionia, *Tenacibaculum*, and *Salipiger* species, including *Salipiger* sp., whereas AHPND-infected shrimp showed higher abundance of Caulobacterales, Cytophagales, Chitinophagales, Saprospiraceae, Cyclobacteriaceae, *Lewinella*, and *Nautella* taxa (Figure 2B). At 18 dph, healthy shrimp exhibited enrichments of Bdellovibrionia, Bacteroidales, *Tenacibaculum*, Alteromonadales, and *Salipiger* species, including *Salipiger* sp. In contrast, AHPND-infected shrimp had increased abundances of Bacteroidota, Bacteroidia, Caulobacterales, Saprospiraceae, Sneathiellales, Sneathiellaceae, Sneathiella, Phaeodactylibacter, Neptunomonas, *Sneathiella* sp., and *Phaeodactylibacter* sp. (Figure 2C). Conversely, there were almost no differential taxa identified between healthy and AHPND-infected shrimp at 21 dph (Figure 2D).

### 3.4. Ecological Processes Governing the Postlarval-Associated Microbiota

In comparison to healthy controls, AHPND-infected shrimp demonstrated a greater reliance on stochastic processes, with increased contributions of drift (40.5% in AHPND postlarvae vs. 31.6% in healthy controls) and dispersal limitation (44.7% vs. 32.7%) governing their associated bacterial communities, with the notably compromised importance of homogeneous selection (35.0% vs. 14.3%) (Figure 3A).

To quantify the contribution of each phylogenetic lineage to various ecological processes, ASVs were grouped into 66 phylogenetic lineages (Appendix A). Notably, only four phylogenetic lineages accounted for the differences observed in homogeneous selection, dispersal limitation, and drift between healthy and infected shrimp (Figure 3B), despite each bin containing a limited number of ASVs (Appendix A). Bin36 Rhodobacteraceae species increased stochasticity, leading to an increase in drift (18.1%) and dispersal limitation (14.6%) in AHPND-infected shrimp, compared to their contributions of 11.9% and 3.0% in healthy shrimp, respectively. Conversely, these species resulted in reduced homogeneous selection (8.1% in AHPND vs. 21.2% in healthy shrimp) in the AHPND-infected group (Figure 3B). Similarly, Bin63 *Vibrio* species enhanced drift (2.7% vs. 5.6%) but decreased homogeneous selection (0.4% vs. 2.8%) and dispersal limitation (14.6% vs. 3.0%) in AHPND-infected shrimp (Figure 3B). In contrast, Bin11 Flavobacteriaceae taxa increased dispersal limitation (7.2% vs. 0.7%), whereas Bin9 *Bacillus* members reduced homogeneous selection in AHPND-infected shrimp (1.7% vs. 7.9%) compared to healthy controls (Figure 3B).

### 3.5. Identifying AHPND-Discriminatory Taxa Diagnosing Shrimp Health Status

The top 19 dph-discriminatory ASVs accounted for 65.6% of the variance related to dph (Appendix A) and were excluded to eliminate variations in the bacterial community during the ontogeny of healthy shrimp. Subsequently, the AHPND-discriminatory taxa were identified by stratifying healthy and AHPND cohorts. The optimal combination of the top 13 infection-discriminatory taxa was determined using 10-fold cross-validation. These biomarkers were predominantly associated with Alphaproteobacteria, Bacilli, Gammaproteobacteria, and Bacteroidia populations. Additionally, detrimental ASV328 *Bartonella* sp., ASV193 *Nannocystis* sp., ASV244 *Membranihabitans* sp., ASV364 *Longitalea* sp., and ASV444 *Spongiibacter* sp. were significantly enriched in AHPND-infected postlarvae compared to their healthy counterparts. Conversely, beneficial ASV165 *Phyllobacterium* sp., ASV34 *Salipiger* sp., ASV560 *Parahaliea* sp., ASV483 *Clostridium* sp., ASV325 *Delftia* sp., ASV19 *Bacillus* sp., ASV8 *Bacillus* sp., and ASV58 *Bacillus* sp. demonstrated the opposite trend (Figure 4B). Employing the profiles of the 13 biomarkers and their relative importance as dependent variables, the model was utilized to diagnose the health status of each sample. Intriguingly, we achieved an overall 100% accuracy in stratifying shrimp health status (Figure 4C).

### 3.6. Interspecies Interaction Between Healthy and AHPND Postlarvae

To explore how interspecies interactions among commensals were affected by AHPND, we divided the bacterial communities into two datasets: healthy and AHPND-infected cohorts, following the removal of the 19 dph-discriminatory ASVs (Appendix A). The resulting two networks exhibited significant differences in terms of nodes and topological features (Figure 5). Specifically, the AHPND-infected network was less complex than the healthy network, characterized by fewer edges (625 compared to 849 in the healthy network), nodes (169 compared to 180), average degree (7.39 versus 9.43), and average clustering coefficient (0.379 versus 0.409) (Appendix A). Accordingly, AHPND-infected shrimp demonstrated lower network stability than healthy controls, as evidenced by reduced total cohesion and robustness, along with increased vulnerability (Figure 5D–F). Thus, AHPND infection results in a less competitive, more fragmented, and vulnerable network, ultimately compromising network stability.

To further identify the key taxa driving the shift from a healthy to an AHPND-infected status, we calculated the NetMoss score for each node. This analysis revealed 18 ASVs categorized as keystone taxa, with NetMoss scores greater than 0.8 and a Wilcoxon test *p*  < 0.05 (Figure 5F). Among these, the relative abundances of 12 ASVs were enriched in healthy postlarvae and were positively correlated with network robustness. In contrast, two ASVs (ASV5 *Gilvibacter* sp. and ASV99 *Lewinella* sp.) that were enriched in AHPND-infected postlarvae displayed a negative association with network robustness (Figure 5F). These keystone ASVs are likely implicated in AHPND pathogenesis.

## 4. Discussion

The intricate interplay between host-associated microbiota and health has attracted significant attention in recent years [5,40,53]. *P. vannamei* postlarvae face severe threats from AHPND, a devastating disease that leads to rapid and massive mortality [10]. Therefore, we investigate the dynamics of postlarvae-associated microbiota during the progression of AHPND, particularly focusing on the driving taxa and underlying processes, which enhance our understanding of AHPND etiology from an ecological perspective.

There is evidence that microbiota profiles can vary depending on the selected region of the 16S rRNA gene for amplicon sequencing [54]. Here, we used the most widely deployed V3–V4 hyper-variable regions of bacterial 16S rRNA genes to improve resolution. The dominant bacterial phyla (Appendix A) and genera (Appendix A) have been reported in relevant works [25]. In addition, we focused on divergence in postlarvae-associated microbiota over AHPND progression, rather than compared taxa level with other studies. In this regard, the selection of hyper-variable regions could not strongly affect our conclusion. The microbiota is not exclusively focused on the primary target organ of hepatopancreas. However, the microbiotas in other tissues are also important in the outcome of host health. Therefore, the exploration of postlarvae-associated microbiota in AHPND pathogenesis is still valuable.

It is widely accepted that higher bacterial diversity contributes to a more stable and resilient community in the face of disturbances [55]. Consequently, bacterial α-diversity is regarded as a crucial indicator of community stability [5]. Our findings indicate that the diversity of AHPND-infected postlarvae was significantly lower than that of matched healthy controls at 11 dph, but this was not observed in advanced AHPND cases (Figure 1A). A possible explanation for this is that pathogen infection outcompetes postlarval-associated symbionts upon the onset of disease. Alternatively, PirAB induces massive epithelial sloughing. The resulting loss of tissue could reduce the available substrate for postlarval-associated bacteria, thereby contributing to a reduction in bacterial community at AHPND onset stage. However, as the disease progresses, the selection for alien species may diminish, leading to random colonization by these species thereafter [33]. This resulted in only a transient reduction in α-diversity (Figure 1A). Consistent with this assertion, we observed that the influence of homologous selection was suppressed, whereas the impact of dispersal limitation was enhanced in AHPND-infected shrimp compared to healthy postlarvae (Figure 3). Also, opportunistic bacterial colonization of the hepatopancreas occurs after the epithelium has been sloughed and the tissue has become necrotic, resulting in the recovery of bacterial diversity as AHPND progressed. In contrast, the gut microbial α-diversity was comparable between healthy and AHPND-infected shrimp at the onset of the disease, but it was significantly lower in AHPND-advanced adults [11,22]. The structure of the initial microbiota plays a key role in shaping responses to disturbances [26]. By this logic, this discrepancy may be explained by differences in the microbial communities associated with postlarval and adult shrimp. Moreover, adult shrimp surviving AHPND enter a chronic disease state characterized by severe hepatopancreatic degradation, including hemocytic infiltration, bacterial encapsulation, and tissue atrophy [56]. These pathological changes create a microenvironment that restricts bacterial colonization, thereby reducing microbial diversity in chronic cases. Notably, significant differences in the beta-diversity of microbiotas were detected based on health status at 11, 15, and 18 dph, but not at 21 dph (analysis of similarity *r* = 0.072, *p* = 0.249, Table 1, Appendix A). Shrimp mortality occurred at 11 dph, following the emergence of disease symptoms; however, postlarval mortality ceased by 21 dph (Appendix A). In accordance, the recovery of the bacterial community positively influenced shrimp survival rates under *V. parahaemolyticus* challenge [57]. The correlation between shrimp mortality and dysbiosis in the postlarval-associated microbiota highlights the critical role that microbial symbionts play in maintaining host health. However, variations in shrimp age led to limited consistency in observed microbial differences between healthy and AHPND-infected postlarvae across time points, with only minor distinctions in dominant genera between the groups (Figure 2), indicating the high temporal dynamics of postlarvae-associated microbiota. In this regard, the exploration of postlarvae AHPND etiology should condition shrimp dph.

There is evidence that gut Firmicutes and Bacteroidota species serve as key commensals in host fermentation and nutrient provision [58,59]. Specifically, gut Firmicutes species enhance lipid metabolism in shrimp, improving energy harvest [60], whereas Bacteroidota members contribute to carbohydrate metabolism, thereby enhancing overall energy metabolism [4]. Following this reasoning, the observed enrichment of gut Bacteroidota and the suppression of Firmicutes in AHPND-affected shrimp (Figure 3 and Appendix A) may promote carbohydrate metabolism but weaken lipid metabolism. Consistently, AHPND-infected shrimp exhibit suppressed fatty acid metabolism and increased carbohydrate metabolism [61,62]. Furthermore, Firmicutes/Bacteroidetes ratio in AHPND-affected shrimp (0.21 ± 0.23) was significantly lower than that in healthy postlarvae (0.89 ± 0.46), mirroring dysbiotic patterns linked to inflammatory diseases in other shrimp species [63,64]. The Firmicutes/Bacteroidetes ratio is well recognized for its critical role in maintaining normal gut homeostasis, which potentiates shrimp’s capacity to mitigate oxidative stress or repair PirAB toxin-induced tissue damage. Thus, substantial alterations in Firmicutes/Bacteroidetes ratio significantly impact shrimp energy metabolism, consequently increasing shrimp susceptibility to AHPND, aligning with prior reports that a decreased ratio induce acute inflammatory diseases [63].

Owing to the rapid exacerbation and high mortality associated with shrimp infected by AHPND [10], our objective was to develop a diagnostic model for detecting AHPND infection status. In addition to the notable differences in microbiota based on shrimp health status, shrimp age (measured in days post hatching) significantly influenced the postlarval-associated microbiota (Figure 1B), reflecting similar findings in adult shrimp gut microbiota [65,66]. The 19 dph-discriminatory ASVs constrained 65.6% of the observed microbial variation (Appendix A). The remaining unexplained variation could be attributed to other unmeasured/unenrolled abiotic and biotic factors, such as changing water geochemical variables and matured immunity as shrimp age. To reduce the influence of dph on the bacterial community, we identified 13 AHPND-discriminatory ASVs after removing the top 19 dph-discriminatory ASVs from the bacterial community. The biomarkers accurately distinguished AHPND-infected postlarvae from healthy individuals with 100% diagnostic accuracy (Figure 4). Notably, the functional relevance of these biomarkers can be further evaluated based on their biological and ecological roles. For instance, the abundance of ASV193 *Nannocystis* sp. in AHPND-infected shrimp was significantly higher than in healthy postlarvae (Figure 4B). *Nannocystis* sp., a member of Myxococcota, is known to cause diseases in both animals and humans by inducing local hypoxia, which leads to oxidative injury and disrupts the homeostasis of the gut microbiota [67]. Conversely, the relative abundances of ASV483 *Clostridium* sp. and three *Bacillus* species were significantly reduced in AHPND-affected shrimp compared to healthy controls (Figure 4B). *Clostridium* species are well-established probiotics in aquaculture, which produce butyric acid to suppress the virulence of harmful factors like the PirAB toxin and modulate anti-inflammatory responses [68]. Although AHPND is directly caused by extracellular PirAB toxin, AHPND-causing pathogens are frequently isolated from the gut of diseased shrimp [69]. Thus, the reduced abundance of *Clostridium* sp. may compromise shrimp immune defenses and their ability to resist colonization by pathogens. Additionally, *Bacillus* strains are known to degrade macronutrients and enhance intestinal digestibility by secreting digestive enzymes in shrimp [70], and they also have the capacity to inhibit the growth and spread of pathogens [71]. In this context, the suppressed biomarkers in AHPND-infected cohorts may serve as candidate probiotics for enhancing disease resistance. Importantly, our diagnostic model was able to accurately identify the preclinical stage of AHPND, including diseased individuals at 11 dph (Figure 4C); however, the practical application of this approach necessitates further validation. In addition, given the lack of strain level primer, it is challenging to detect these biomarkers using qPCR. Nevertheless, with the rapid decline in sequencing price, this approach might be used at fisheries technology extension stations by skilled technicians.

To better understand how progressed AHPND impacts the ecological processes that regulate the microbiota associated with postlarvae, we compared the significance of these processes in relation to shrimp health status at each disease stage (Figure 3). The importance of homogeneous selection in AHPND-infected postlarvae was consistently and significantly diminished compared to matched healthy controls (Figure 3A). Similarly, the influence of homogenizing selection was reduced in adult shrimp affected by AHPND [22]. From an ecological perspective, weakened homogeneous selection leads to the random colonization of external species, resulting in divergences within the microbiota [43]. In line with this, AHPND-infected shrimp displayed increased significance of dispersal limitation and drift, which together contribute to more random variations in community structure (Figure 3A). These shifts in underlying ecological processes may create opportunities for pathogen invasion [33], thereby exacerbating the progression of AHPND, as observed here and in other studies [22,72].

Notably, shifts in ecological processes were triggered by a select few bacterial lineages (Figure 3B). Rhodobacteraceae symbionts play a beneficial role in enhancing nutrient absorption and utilizing complex organic compounds in shrimp, fostering a stable microbiome [34]. However, our findings indicate that Bin36 Rhodobacteraceae species were enriched, whereas Bin63 *Vibrio* taxa decreased in the AHPND network compared to the healthy counterpart (Figure 3B and Appendix A). A microbial lineage that differentially affects ecological processes has not been reported, warranting further investigation to reveal the underlying mechanisms. Bin36 Rhodobacteraceae species promoted homogenous selection in healthy shrimp, whereas they induced dispersal limitation and drift in AHPND-infected postlarvae (Figure 3B). In accordance with this, the Rhodobacteraceae population has a disproportionate role in maintaining network stability, which was positively correlated with the up-regulation of amino acid metabolism and NF-κB signaling pathways [73]. It has been elucidated that external disturbances stimulate the immigration of Rhodobacteraceae species from rearing water into the shrimp gut [74]. In this regard, the enriched Rhodobacteraceae in AHPND-affected postlarvae could be an adaptation strategy, and future work should aim to unveil the underlying mechanisms. In addition to opportunistic pathogens, *Vibrio* species also serve as important gut symbionts, conferring diverse metabolic activities to shrimp [23]. The enriched Bin63 *Vibrio* in healthy shrimp primarily contributed to dispersal limitation, followed by homogenous selection (Figure 3B). It has been proposed that when host selection acts on the microbiota, dispersal limitation may be influenced by deterministic processes, as dispersal rates depend on interactions with the resident community and the host [75]. In this context, the enriched Bin63 *Vibrio* could enhance network stability. Thus, the biological roles of the *Vibrio* genus should be interpreted with caution. Flavobacteriaceae represents a dominant population in the postlarval shrimp-associated microbiota [76]. The enrichment of Flavobacteriaceae has the potential to promote dispersal limitation [77], a pattern that aligns with our findings (Figure 3B, see Flavobacteriaceae members in Appendix A). *Bacillus* species are widely recognized for their probiotic attributes, which significantly enhance shrimp growth performance and health [78]. Consistent with this, *Bacillus* species exhibited a significantly greater prevalence (*p* < 0.001) in healthy shrimp compared to the AHPND cohort, thereby sustaining homogenous selection in healthy individuals (Figure 3B). Overall, AHPND undermines the active selection of ambient species in shrimp, a change triggered by specific bacterial bins. Understanding these dynamics is crucial for developing strategies to mitigate the detrimental impacts of AHPND on shrimp aquaculture, such as the supplementation of Bin36 Rhodobacteraceae species and Bin9 *Bacillus* species.

The co-occurrence network provides valuable insights into community structure and assembly patterns, with its complexity serving as an index of community stability and resilience to disturbances [48]. Our observations indicate that healthy shrimp exhibit a more complex, better-connected, and highly cooperative network compared to their AHPND-infected counterparts (Figure 5). A complex network enhances the ability to resist pathogen invasions and environmental disturbances, thereby increasing the stability of the microbiota [79]. In contrast, a less cooperative community is more vulnerable to trophic shifts, which undermines its resilience to disturbances [80]. Consistent with this, we found that the occurrence of AHPND disrupted cooperative activities among shrimp symbionts and compromised network robustness (Appendix A and Figure 5), subsequently diminishing their active selection for the colonization of external strains (Figure 3). The negative correlations observed between microorganisms in AHPND-infected shrimp may stem from direct inhibition, toxin production, and competition [81]. Furthermore, AHPND stress reduces network invulnerability, which corresponds with a decline in the abundance of keystone taxa (Figure 5F,G). Of the 18 keystone ASVs identified as responsible for the transition from a healthy to a diseased network, most were found to be diminished in AHPND-infected shrimp and were significantly associated with network robustness (Figure 5G). Conversely, a few keystone ASVs, such as ASV5 *Gilvibacter* sp. and ASV99 *Lewinella* sp., were enriched in AHPND-infected shrimp. In accordance with this, it has been outlined that the AHPND infection caused by *V. parahaemolyticus* promotes the proliferation of gut *Gilvibacter* and *Lewinella* species [25,82]. Collectively, our findings indicate that the occurrence of AHPND leads to a reduction in the abundance of keystone symbionts, ultimately resulting in decreased network stability.

## 5. Conclusions

AHPND significantly alters the structural composition of the postlarval shrimp microbiota, leading to a marked imbalance among dominant lineages, such as decreased Firmicutes/Bacteroidetes ratio. However, the bacterial communities are comparable between healthy and AHPND-infected cohorts at the later disease stage, which is accompanied with halting death. Shrimp AHPND compromises the importance of homogenous selection on their associated microbiota, which is primarily driven by Bin36 Rhodobacteraceae, Bin11 Flavobacteriaceae, Bin63 *Vibrio*, and Bin9 *Bacillus* species. However, a given bin divergently affects the ecological processes between postlarvae health status, whereas the underlying mechanism merits further exploration. Additionally, AHPND-infected postlarvae demonstrate reduced network stability, compromised robustness, decreased connectivity, and diminished positive cohesion, accompanied by the loss of keystone species. Notably, we attempt to develop a diagnostic model that can accurately differentiate healthy shrimp from AHPND-infected individuals. However, further validation of this promising method is necessary before it can be implemented in practice. Collectively, these findings narrow the knowledge gap on the inextricable interrelationship between postlarval shrimp health, their associated microbiota, and survival, as well as the underlying ecological mechanisms at play.

## Figures and Tables

**Figure 1 microorganisms-13-00720-f001:**
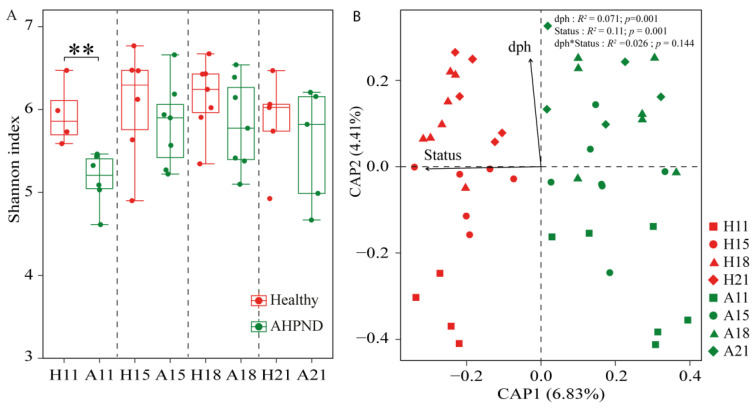
The diversity and structures of bacterial community in healthy and acute hepatopancreatic necrosis disease (AHPND)-infected postlarval shrimp. (**A**) Comparison of Shannon index between healthy and AHPND cohorts at each sampling using Wilcoxon rank-sum test. **: *p* < 0.01. (**B**) Constrained analysis of principal coordinates revealing the effects of days post hatching (dph) and AHPND on the larvae-associated microbiota. PERMANOVA evaluates the importance of dph and healthy status in shaping bacterial community structure. H: healthy shrimp; A: AHPND-infected shrimp. Numbers are dph. dph*status indicates the interactive effect of dph and shrimp health status on the variance in postlarval shrimp-associated microbiota.

**Figure 2 microorganisms-13-00720-f002:**
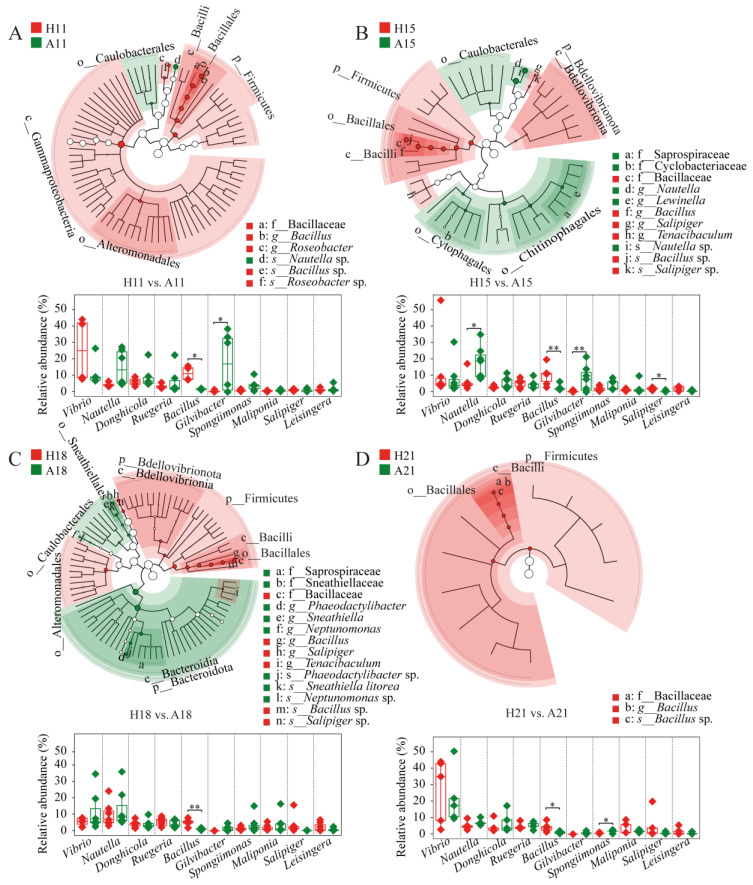
The differential taxa and relative abundance of the 10 most abundant genera between AHPND-infected larvae and matched healthy shrimp at (**A**) 11, (**B**) 15, (**C**) 18, and (**D**) 21 days post hatching using LEfSe analysis and unpaired *t*-test, respectively. **: *p* < 0.01 and *: *p* < 0.05. The circle diameter is proportional to the abundance of a bacterial taxon. Red and green indicate the differential taxa that are enriched in healthy and AHPND-infected larvae, respectively. Refer to Table 1 for abbreviations.

**Figure 3 microorganisms-13-00720-f003:**
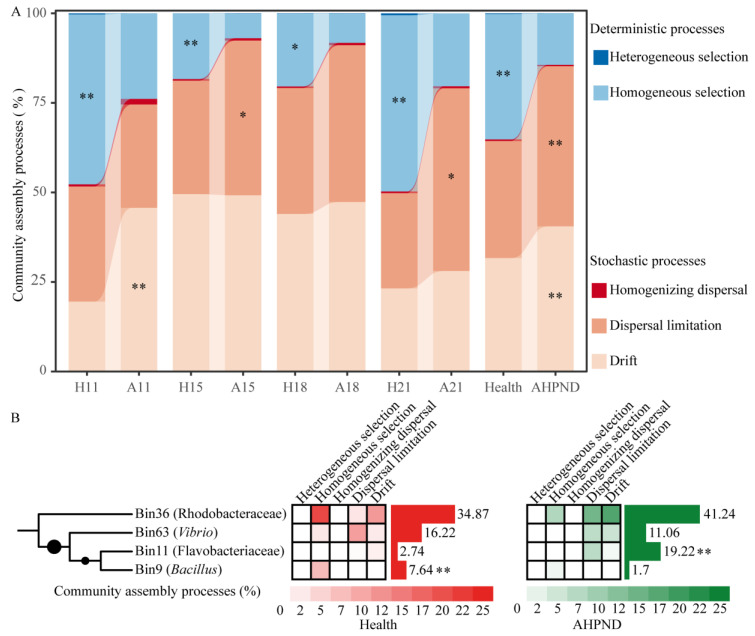
AHPND-induced shifts in microbial community assembly dynamics. (**A**) Contrasting assembly processes between shrimp health statuses at each sampling using Wilcoxon rank-sum test. *: *p* < 0.05; **: *p* < 0.01. (**B**) Phylogenetic bin-based null model analysis identifies the important bins triggering the ecological processes of bacterial communities. Relative abundances of the bins between healthy and AHPND cohorts are compared using Wilcoxon rank-sum test. **: *p* < 0.01. Refer to Table 1 for abbreviations.

**Figure 4 microorganisms-13-00720-f004:**
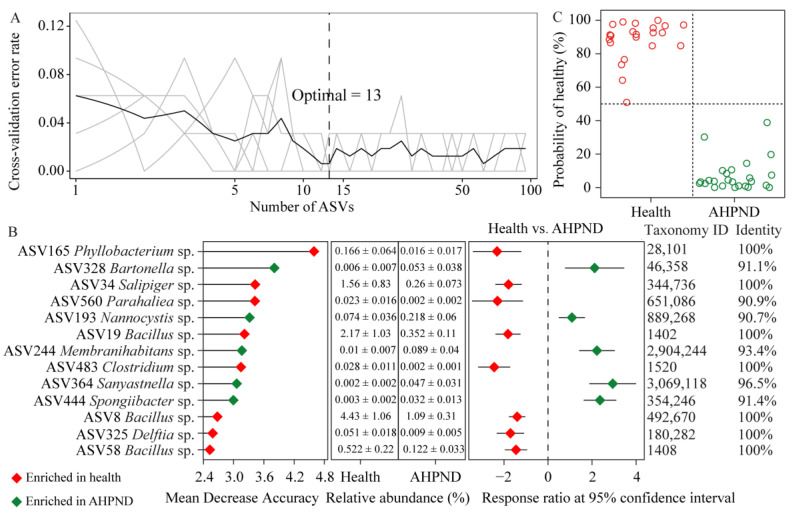
Identification of biomarkers for diagnosing AHPND infection. (**A**) The top 13 AHPND-discriminatory ASVs are ascertained by 10-fold cross-validation. (**B**) The importance scores of AHPND-discriminatory ASVs and their differences between larval shrimp health status using response ratio analysis at a 95% confidence interval. (**C**) The predicted and diagnosed health status with a cutoff of 50%.

**Figure 5 microorganisms-13-00720-f005:**
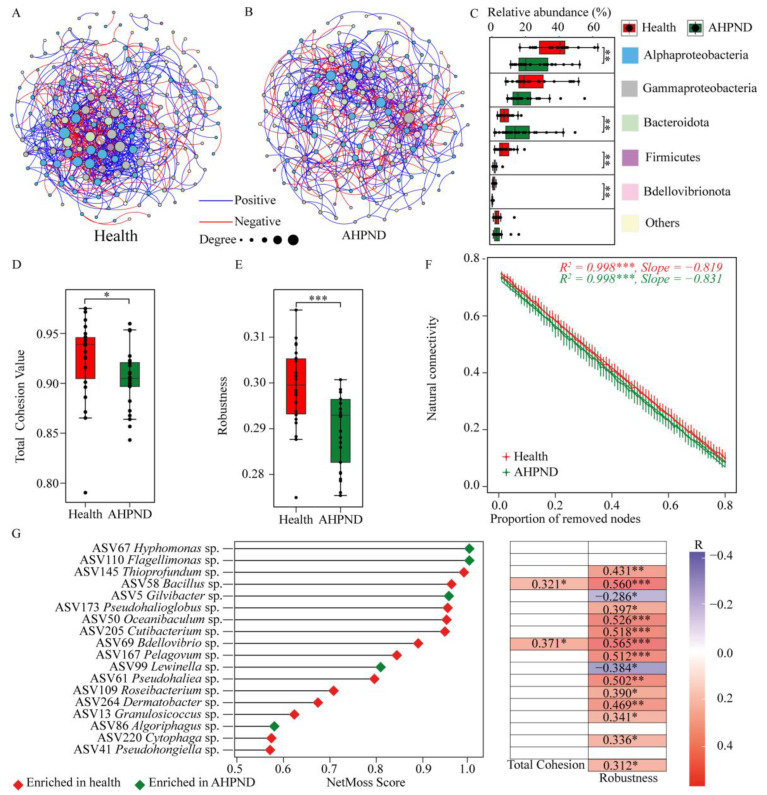
Comparison of networks and their properties between healthy and AHPND-infected shrimp after removing the dph-discriminatory taxa. Co-occurrence networks for (**A**) healthy and (**B**) AHPND-infected shrimp. Comparison of (**C**) relative abundances of the network nodes that are categorized into bacterial phylum level, (**D**) network total cohesion, (**E**) robustness, and (**F**) connectivity (invulnerability) between healthy and AHPND cohorts using Wilcoxon rank-sum test. ***: *p* < 0.001; **: *p* < 0.01 and *: *p* < 0.05. (**G**) Keystone nodes that trigger shifts in the network from health to AHPND status, and the associations between their relative abundance and network characteristics using the Spearman correlation.

**Table 1 microorganisms-13-00720-t001:** Analysis of similarity of the postlarvae-associated bacterial communities between groups using Bray–Curtis dissimilarity. Top diagonal cells are *r* values, and lower diagonal cells are *p* values. H: healthy shrimp; A: AHPND-infected shrimp. Numbers are days post hatching.

	Healthy				AHPND			
	H11	H15	H18	H21	A11	A15	A18	A21
H11	-	0.179	0.524	0.213	0.528	0.836	0.817	0.55
H15	0.159	-	−0.021	0.133	0.631	0.487	0.431	0.365
H18	0.004	0.491	-	0.144	0.609	0.457	0.272	0.344
H21	0.128	0.155	0.127	-	0.581	0.642	0.355	0.072
A11	0.012	0.004	0.001	0.002	-	0.139	0.447	0.323
A15	0.004	0.002	0.001	0.002	0.115	-	0.082	0.464
A18	0.006	0.002	0.011	0.024	0.004	0.21	-	0.025
A21	0.016	0.015	0.022	0.249	0.027	0.005	0.335	-

## Data Availability

The original contributions presented in this study are included in the article/Appendix A. Further inquiries can be directed to the corresponding authors.

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
