# Peer review of "Postlarval Shrimp-Associated Microbiota and Underlying Ecological Processes over AHPND Progression"

_microorganisms, 2025, doi:10.3390/microorganisms13040720_

Round 1
Reviewer 1 Report
Comments and Suggestions for Authors
Dear authors,
The study is interesting, but I am concerned about the experimental designs. First, you do not ensure that the AHPND larvae were affected by the disease. A PCR and histology analysis are necessary. What were the clinical signs of larvae AHPND? The discussion does not explain your results, so the conclusions are not good.

Reviewer 2 Report
Comments and Suggestions for Authors
Review of the Manuscript: “Larval shrimp-associated microbiota and underlying ecological processes over AHPND progression”
The manuscript entitled “Larval shrimp-associated microbiota and underlying ecological processes over AHPND progression” presents an insightful study on the bacterial microbiota associated with both AHPND-infected and healthy shrimp. Through amplicon sequencing of the V3-V4 region and bioinformatics analyses, the authors identified 13 AHPND-discriminatory taxa and developed a diagnostic model capable of distinguishing healthy from infected individuals with 100% precision. The study is well-executed and holds merit for publication; however, I recommend acceptance with major revisions.
The authors must explicitly acknowledge that, while analyzing the whole organism (a reasonable approach given the small size of the shrimp), their study does not focus exclusively on the microbiota of the hepatopancreas—the primary target organ of the AHPND toxin. Consequently, their dataset includes bacterial communities from other tissues such as the gills and intestines, which may not have a direct role in AHPND pathogenesis.
The discussion must be revised to ensure accurate interpretation of AHPND. The authors must recognize that AHPND is not an infection but rather an intoxication caused by the binary toxin PirAB, which leads to hepatopancreatic destruction.
The authors should include a dedicated paragraph in the discussion clarifying that microbiota profiles can vary depending on the selected region of the 16S rRNA gene for amplicon sequencing. This variability should be explicitly acknowledged, as different primer sets may yield distinct taxonomic compositions.
Specific Comments:
Lines 10-12: The text should clarify that acute hepatopancreatic necrosis disease (AHPND) is the formal name of the disease, whereas early mortality syndrome (EMS) was a provisional term used before the etiological agent was identified. Since EMS is now outdated, the manuscript should exclusively use AHPND.
Line 35: The correct taxonomic name is Penaeus vannamei, and it should be used consistently throughout the manuscript.
Line 40: Replace all occurrences of "EMS" with "AHPND." The term "EMS" was applicable only when the causative agent was unknown and should no longer be used in scientific literature.
Lines 70-71: The statement should be affirmative—larval and adult shrimp exhibit distinct physiological responses to AHPND, and this should be stated unequivocally.
Lines 132-135, 213-224, and 245-262: Key findings from these sections should be summarized in the abstract to improve the manuscript’s clarity and accessibility.
Lines 296-298: The discussion should explicitly state that only 65% of the variation was explained by discriminatory ASVs. The remaining unexplained variation should be acknowledged, as it may still contribute to microbiota shifts during AHPND progression.
Lines 359-364: The authors must recognize that AHPND results from intoxication by the PirAB toxin rather than bacterial infection. There is no conclusive evidence that Vibrio species producing PirAB colonize the hepatopancreas. Therefore, the argument that Vibrio outcompetes the larval-associated microbiota in the hepatopancreas is not valid. Furthermore, as PirAB induces massive epithelial sloughing, the resulting loss of tissue could reduce available substrate for larval-associated bacteria, contributing to microbial shifts. The potential direct effects of PirAB on non-Vibrio bacterial taxa should also be considered. Finally, the authors should acknowledge that opportunistic bacterial colonization of the hepatopancreas generally occurs after the epithelium has been sloughed and the tissue has become necrotic.
Lines 367-369: The authors should note that shrimp that survive AHPND into adulthood enter a chronic disease state (Aranguren et al., 2020). These individuals exhibit a severely compromised hepatopancreas with extensive hemocytic infiltration, encapsulated bacteria, and significant atrophy—conditions that severely limit potential bacterial colonization.
Line 374: The term symptoms should be replaced with signs, as "symptoms" apply to subjective experiences in humans, whereas "signs" refer to observable clinical manifestations in animals.
This study provides valuable insights into the microbiota dynamics associated with AHPND in shrimp larvae. However, the authors must address the critical misinterpretations regarding AHPND pathogenesis and ensure that their findings are framed appropriately within the established understanding of the disease. Additionally, key methodological limitations—such as the inclusion of microbiota from non-relevant tissues and the inherent variability introduced by 16S rRNA sequencing regions—must be explicitly discussed. Addressing these concerns will significantly improve the manuscript’s scientific rigor and impact.
Reference
Fernando, L., Caro, A., Mai, H.N., Noble, B., Dhar, A.K., 2020. Acute hepatopancreatic necrosis disease ( VP AHPND ), a chronic disease in shrimp Penaeus vannamei population raised in latin America. J. Invertebr. Pathol. 174, 107424. https://doi.org/10.1016/j.jip.2020.107424
Reviewer 3 Report
Comments and Suggestions for Authors
In this study, the authors have investigated the microbiota associated with healthy and AHPND-diseased shrimp using molecular and biostatistical approaches in the aim to develop a diagnostic model based on 13 biomakers. They also demonstrated that the microbiota is related to the larval health, survival and day after hatching, and interact with the host health.
The manuscript is clear and well organized, cite appropriate references and use strong statistical analysis of the data that are well described tin the material and methods sections as well as in the results part.
After carefully reading this manuscript, I have few suggestions and comments as follows: that could help to strength the manuscript:
*All the taxonomic names through the entire manuscript should be in italic, at least in the text (see: Oren, A., Arahal, D. R., Göker, M., Moore, E. R., Rossello-Mora, R., & Sutcliffe, I. C. (Eds.). (2023). International code of nomenclature of prokaryotes. Prokaryotic code (2022 revision). International Journal of Systematic and Evolutionary Microbiology, 73(5a), 005585).
*Details are needed in the Experimental Design and Sampling section: is it the whole shrimp, part of the shrimp (if yes, which one) or the gut that was sampled?
*Throughout the whole manuscript there are a lot of typo with missing space between 2 words or between the word and the reference such as cultivation[2] or sequencingdepth
*Can you explain how the healthy shrimps are the control? Have you looked for diseased in these rearing tanks? Which parameters allow to say that they are really healthy?
*At 11 days post hatching, Litopenaeus vannamei should be in postlarvae stage, at least at 15dph they should be postlarvae ? If not, can you explain why there is a delay in the growth phase; otherwise, change larvae by post larvae in the manuscript.
*Can't go to the species level using metabarcoding, please replace all the species name by sp or lineage affiliated to the genus XX for example, line 260 you should write Phaeodactylibacter sp or members related to the Phaeodactylibacter genus instead of Phaeodactylibacter luteus. Please revise through the whole manuscript
*A Figure that compared the relative abundance of the dominant bacterial genera between AHPND infected shrimp using unpaired t test, should be added in the main text. That allows the readers to see the proportion/relative abundance of the main genera and to relate this figure with the LEfSe figure (Figure 2) .
* A sentence about the method used should be added in the abstract, such as "To address this knowledge gap, a comparative analysis of larvae-associated microbiota and the ecological processes underlying AHPND progression was performed by sequencing of the bacterial V3-V4 hypervariable region of the 16S rRNA gene."
*In the section 2.3. Amplicon Data Processing: which method the authors used to remove the chimera? and Why did you choose to use the rarefaction method for the data normalization? Indeed the highest library encompasses 114,866 reads while the lowest has 77,110 reads meaning that with the rarefaction the highest library loose half of its reads. Why not using the TSS method that divided the number of total reads of each sample, or the CSS
*The LEfSe data should be added in the discussion, especially in the paragraph starting at the line 440
*Line 424 and In the conclusion, the author wrote about the development of a diagnostic model that can accurately differentiate healthy shrimp from AHPND-infected individuals, achieving an overall accuracy of 100%. This diagnostic model should be more discussed especially about the detection of these 13 biomarkers in shrimps: used of molecular technic? Are they found in the early stage of the disease? Need to collect shrimp and to perform DNA extraction and PCR? Sequencing? How that can be used as monitoring tool or bio-surveillance tool?
*Line 37: artificial reproduction instead of Artificial propagation
*Line 120: L. vannamei larvae were collected instead of L. vannamei larvae were enrolled
*Line "At the 11 days post hatching (dph), mortality was observed in a few tanks, with the presence of detectable pirAB genes in the affected shrimp." Are the data available if yes should be added in Supp info along with the sequences or references of the primers pirAB
Reviewer 4 Report
Comments and Suggestions for Authors
This study investigates the microbiota dynamics of Litopenaeus vannamei larvae during Acute Hepatopancreatic Necrosis Disease (AHPND) progression. The research provides valuable insights into the ecological mechanisms driving microbiota changes in response to disease and identifies potential biomarkers for early detection of AHPND. The study is well-structured, methodologically sound, and contributes significantly to shrimp disease ecology. However, there are some areas where the manuscript could be improved in terms of clarity, statistical robustness, and discussion depth.
- The discussion effectively interprets the results in the context of microbial ecology, but additional references to previous AHPND microbiome studies would strengthen the argument.
- The Firmicutes/Bacteroidota ratio is linked to metabolic shifts, but the mechanism connecting these shifts to shrimp mortality could be discussed more explicitly.
- The study suggests that Bin36 Rhodobacteraceae and Bin63 Vibrio have opposite effects on community stability, but their role in host-pathogen interactions remains speculative. More mechanistic explanations are needed.
- Figures are generally well-presented, but some legends lack clarity. For example, Figure 3 (ecological process shifts) should specify how each bin was identified and whether it was statistically significant.
- The network analysis visualization (Figure 5) is informative but could be improved by using node size variations to reflect the importance of keystone taxa.
- Line 67-70: The claim that larval and adult shrimp respond differently to AHPND should be supported with additional citations.
- Line 155-165: The process of removing dph-discriminatory ASVs is crucial but not explained well. Did this significantly alter the overall diversity metrics?
- Line 368-372: The statement on microbiota recovery at 21 dph could be supplemented with statistical tests to confirm if differences were non-significant.
- Figure 4B: The relative abundance differences of biomarkers should be displayed with confidence intervals to indicate variability.
- Provide a detailed justification for the denoising parameters and microbial filtering criteria used in QIIME2.
- Clarify whether the machine learning model was externally validated to confirm its generalizability.
- Expand the discussion on mechanistic links between microbial shifts and shrimp mortality.
- Improve figure legends and data visualization, particularly in network analysis and biomarker identification.
- Include additional references on AHPND microbiota research to strengthen the discussion.
Round 2
Reviewer 3 Report
Comments and Suggestions for Authors
Numerous improvement of the manuscript have been done which make clearer the study. The authors have reply to all my comments.
Author Response
The reviewer endorses our efforts in the revision, without further concerns and suggestions.